

**Title**
A novel inter-comparison of nutrient analysis at sea: recommendations to enhance
comparability of open ocean nutrient data
**Authors**
Triona McGrath[1], Margot Cronin[2], Elizabeth Kerrigan[3], Douglas Wallace[3], Clynton Gregory[1],
Claire Normandeau[3], and Evin McGovern[2]
**Affiliations**
1; National University of Ireland, Galway, University Road, Galway, Ireland
2; The Marine Institute, Ireland, Rinville, Oranmore, Galway, Ireland
3; Dalhousie University Canada, LSC Ocean Wing, 1355 Oxford St., PO Box 15000, Halifax,
Nova Scotia, Canada B3H 4R2
**Correspondence Author;**
Triona McGrath; triona_mcgrath@hotmail.com/ triona.mcgrath@marine.ie
Evin McGovern; evin.mcgovern@marine.ie






**Abstract**
An inter-comparison study has been carried out on the analysis of inorganic nutrients at sea
following the operation of two nutrient analysers simultaneously on the GO-SHIP A02 trans-
Atlantic survey in May 2017. Both instruments were Skalar San++ Continuous Flow Analysers, one
from the Marine Institute, Ireland and the other from Dalhousie University, Canada, each
operated by their own laboratory analysts following GO-SHIP guidelines, while adopting their
existing laboratory methods. High quality control of the nutrient analysis was achieved on both
instruments and there was high comparability between the two datasets. Vertical profiles of
nutrients also compared well with those collected in 1997 along the same A02 transect by the
World Ocean Circulation Experiment. The comparison of the two 2017 datasets and individual
laboratory methods, did however raise some interesting questions on the comparison of
nutrients analysed from different systems, in particular the calibration range of daily standards
and its influence on low nutrient samples, and the importance of using certified reference
materials of high and low concentrations to identify bias in the data. Based on the results from
this inter-comparison, a number of recommendations have been suggested that we feel will
enhance the existing GO-SHIP guidelines to improve the comparability of global nutrient datasets.
The A02 nutrient dataset is currently available at the National Oceanographic Data Centre of
Ireland; http://dx.doi.org/10.20393/CE49BC4C-91CC-41B9-A07F-D4E36B18B26F





















## 1. Introduction


Dissolved nutrients such as nitrate, nitrite, silicate and phosphate can be a critical limiting factor
constraining growth of phytoplankton, which in turn form the base of the marine food web. They
also provide useful chemical signatures (e.g. ratios of preformed nutrients) that can distinguish
water masses and their origins (Broecker and Peng, 1982) as well as act as tracers for
biogeochemical processes such as nitrogen fixation and denitrification (Deutsch and Weber,
2012). There is growing evidence for significant variability including long-term trends in nutrient
levels in both coastal (Kim et al., 2011) and open ocean surface (Yasunaka et al., 2014), and deep
waters (Kim et al., 2014). These changes reflect both direct human intervention in the global
environment, especially the effects of the massive ongoing perturbation of the nitrogen cycle
(Yang and Gruber, 2016) as well as changes in ocean circulation and biogeochemical cycling that
may or may not be anthropogenically influenced (e.g. Di Lorenzo et al., 2008).
Identification and attribution of variability of nutrient concentrations has been complicated by
the existence of systematic analytical errors in datasets collected by different groups at different
times. This can lead to controversy over the significance of observed long-term changes (e.g.
Zhang et al., 2001) and generally requires empirical correction of historical data, using a variety
of ad hoc approaches and principles (Keller et al., 2002; Moon et al., 2016; Pahlow and Riebesell,
2000; Tanhua et al., 2009b). Recognition of such systematic errors within and between datasets
led to a series of international comparison studies and the introduction of Certified Reference
Materials for dissolved nutrients (Aoyama et al., 2016; Aoyama et al., 2007), as well as
recommendations concerning standard protocols for sampling, sample preservation and analysis
(Hydes et al., 2010). These steps have undoubtedly contributed to a general improvement in
inter-laboratory comparability of field-collected data. However, it is notable that most inter-
comparison studies rely on either: a) shore-based laboratory-based analysis of replicate samples
in the context of specially organised inter-comparison studies; or b) crossover analysis of
measurements made at nearby locations in the ocean where temporal and spatial variability is
expected to be small.
The former approach is valuable, but most analysts are aware that conditions during an actual
research cruise do not always match the stable, controlled conditions of a shore-based laboratory
where a group can prepare carefully for their measurement of inter-comparison samples. On the
other hand, the latter approach works well in oceanic regions where stable, unchanging nutrient
concentrations can be expected. However, in regions such as the open ocean of the North Atlantic,
or the Northwest Pacific and in coastal regions everywhere, significant "real" temporal and/or
spatial variations can be expected which complicates the interpretation of crossover
comparisons.
In this paper we report the results, findings and lessons learned from a rare opportunity in which
two independent nutrient analysis teams participated jointly in a deep ocean hydrographic
section as part of the international GO-SHIP program (Talley et al., 2016). Both teams followed
standard protocols (Hydes et al., 2010) and both groups used Certified Reference Materials
during the cruise. As such, the cruise provided an opportunity to assess the likely comparability
of nutrient data collected following such protocols as well as helping to identify a number of
issues encountered that could be of general relevance to groups conducting such measurements
elsewhere. We are not aware of any other report of such an extensive, at-sea inter-comparison of
nutrient measurement systems.
The GO-SHIP A02 survey was completed in April/May 2017 on the RV Celtic Explorer, travelling
from St. John's, Newfoundland, Canada, across the North Atlantic to Galway, Ireland with on-
board teams from Ireland, Canada, Germany, the UK, and the USA. The survey provided an





unusual opportunity for cross-comparison of methods, data quality procedures and exchange of
technical expertise between the international scientific groups. The Marine Institute (MI) and
Dalhousie University (Dal) teams brought separate nutrient Skalar San++ auto analysers on the
survey to provide contingency against technical failures and allow for on-board inter-comparison
of data as well as exploration of the impact on data quality of subtle differences in laboratory
methods, procedures and instrument configurations that ostensibly conform to the same (GO-
SHIP) guidelines and quality assurance criteria.

A total of 67 stations were occupied along the A02 transect (Fig. 1), with 1231 nutrient samples
analysed for total oxidised nitrogen (TOxN), nitrite, phosphate and silicate on the MI nutrient
system. Of these, 12 stations were sampled and analysed on the Dal nutrient system, allowing the
comparison of 291 samples between the two systems. The 12 stations were also compared with
historical data from the A02 transect completed on a World Ocean Circulation Experiment survey
in 1997.


## 2. Methods

Sampling, sample preservation and analytical procedures on both systems followed methods
outlined in the GO-SHIP guidelines for nutrient analysis at sea (Hydes et al., 2010), while both
groups also incorporated their existing laboratory quality control (QC), which was specifically
adapted to their individual instruments.

### 2.1 Sampling Procedures

Both groups collected nutrient samples directly from the Niskin bottles into falcon tubes (details
in Table 1) and as per GO-SHIP guidelines, the samples were not filtered. Samples were analysed
on board typically within 12 hours of sampling.

### 2.2 Analytical Methods

Analysis was carried out on two separate Skalar San Continuous Flow Analysers, setup in two
separate on-board containerised laboratories brought by each team. Both nutrient systems run
four channels of nutrients simultaneously; total-oxidised nitrogen, nitrite, silicate and phosphate.
The Dal system also runs ammonia, however there were contamination issues in this channel
during the survey and therefore, there is no further discussion of this method. Both instruments
consist of an auto-sampler, where a needle draws the sample into the analyser which is then spilt
into the four channels. Each channel has its own set of reagents, where the stream of reagents and
samples is pumped through the manifold to undergo treatment such as mixing and heating before
entering a flow cell to be detected. The air-segmented flow promotes mixing of the sample and
prevents contamination between samples. The reagents act to develop a colour which is
measured as an absorbance through a flow cell at a given wavelength. The Skalar Interface
transmits all the data to the Skalar Flow Access software.
The reagents for both systems were made using high-purity chemicals, pre-weighed using a high-
precision calibrated balance prior to the survey. They were stored in acid-washed polyethylene
(PE) containers and mixed to final volume using Milli-Q water, see reagent compositions in Table
161  1.




The analytical procedures for all nutrients are similar between the Dal and MI systems, with some
differences in the chemical composition of reagents and volumes of reagents/sample going
through the instruments (Table 1). For the determination of nitrite, the diazonium compounds
formed by diazotizing of sulfanilamide by nitrite in water under acidic conditions (due to
phosphoric acid in the reagent) is coupled with N-(1-naphthyl) ethylenediamine dihydrochloride
to produce a reddish-purple colour which is measured at 540 nm.

For silicate determination the sample is acidified with sulphuric acid and mixed with an
ammonium heptamolybdate solution forming molybdosilicic acid. This acid is reduced with
L(+)ascorbic acid to a blue dye, which is measured at 810 nm. Oxalic acid is added to avoid
phosphate interference.

For the determination of phosphate, ammonium heptamolybdate and potassium antimony(III)
oxide tartrate react in an acidic medium (with sulphuric acid) with diluted solutions of phosphate
to form an antimony-phospho-molybdate complex. This complex is reduced to an intensely blue-
coloured complex by L(+)ascorbic acid and is measured at 880 nm.

For the determination of total oxidised nitrogen (TOxN) both methods buffer the sample to a pH
of 8.2, which is then passed through a column containing granulated copper-cadmium to reduce
nitrate to nitrite. The nitrite, originally present plus reduced nitrate, is determined by diazotizing
with sulfanilamide and coupling with N-(1-naphthyl) ethylenediamine dihydrochloride to form a
strong reddish-purple dye which is measured at 540nm. The difference between the two systems
is that the MI use a buffer solution made of ammonium chloride and ammonia hydroxide solution,
while the Dal buffer solution is made of imidazole and hydrochloric acid (Table 1). The MI uses a
different Skalar cadmium column where no air bubbles are allowed through the column, while
the Dal system allows air bubbles though their column but monitor the efficiency of the reduction
process and re-activate the cadmium column with 1M hydrochloric acid solution and a copper
sulfate solution if the efficiency falls below 95%.


Both instruments were calibrated daily using a suite of calibration standards (see calibration
range in Table 2). The primary standard for each nutrient was made up in the MI and Dal
laboratories just before the survey using a calibrated balance where the dry weight of each high
purity chemical was diluted to 1L with Milli-Q water, as per Skalar methods. The primary stocks
were stored in the fridge for the duration of the survey. Two batches of primary stocks were used
on the MI system to ensure no bias from an individual batch, while one batch of primary stock
was used on the Dal system. Weekly secondary stocks were made from the primary stocks into
100ml PP flasks which were stored in the fridge when not in use and could be used for one week.
Daily standards were made from secondary stock into 100ml PP volumetric flasks.
The MI secondary and daily calibration standards were made using calibrated fixed volume
pipettes while Dal standards were made using calibrated adjustable volume pipettes (0.1 – 1 ml,
0.5 – 5 ml) and one calibrated fixed volume pipette (10 ml). The adjustable pipettes were tested
prior to the start of the survey to ensure that the volumes delivered were accurate. The MI
secondary stocks were made using Milli-Q water, while the daily standards were made using
artificial seawater (ASW) with salinity of 35. Both secondary and daily standards on the Dal
system were made using ASW (salinity 33-35). Concentrations of daily standards for each system
are in Table 2, where first order calibration was used and $R^2 > 0.99$ was deemed acceptable, as
per Skalar methods.





The MI use ASW as the baseline wash for all channels, at a similar salinity to the expected samples
(salinity 35). Batches of sodium chloride used were tested prior to the survey to ensure no
contamination with any of the nutrients. The Dal system uses Milli-Q water as the baseline wash
and therefore a separate blank is run for each standard curve and set to 0 (e.g. Standard 1 in Table
215    2).


**2.3 Quality Control**

The Certified Reference Materials (CRMs) used on the survey by both groups were supplied from
KANSO (Aoyama et al., 2016; Aoyama et al., 2007) and were analysed at the beginning and end of
every run and monitored daily on quality control charts. Two batches were used (Batch CD and
Batch BW) on the MI system to cover the full range of nutrients expected on the survey, Table 3.
While Dal primarily analysed Batch CD, they analysed a small number of BW CRM as a
comparison.

The nutrient laboratory at the MI is part of a Quality System and participates in the QUASIMEME
laboratory quality control programme where test materials are analysed bi-annually over a large
range of nutrient concentrations and submitted to assess laboratory performance. Since GO-SHIP
guidelines do not give pass/fail criteria for CRMs used during nutrient analysis, CRMs from both
groups were assessed using a z-score criteria as per Quasimeme Proficiency Testing Exercises,
where a z-score < 2 is considered acceptable and z is the difference between the laboratory result
and the certified value, divided by the total error (Cofino and Wells, 1994);
Equation 1;        $z - score = \frac{Measured\ value - Certified\ value}{Total\ error}$

, where the total error is calculated as;

Equation 2;      $Total\ error = \frac{Assigned\ value\ x\ Proportional\ Error\ (6\%)}{100} + 0.5\ x\ Constant\ error$
, and the constant error is 0.05, 0.01, 0.1 and 0.05 µmol/l for TOxN, nitrite, silicate and phosphate,
respectively, which are defined by the Scientific Advisory Board of Quasimeme. Between 2008
and 2017, the average absolute z-scores |Z| from 84 proficiency test samples analysed during
QUASIMEME exercises at the MI laboratory was 0.5 for TOxN, 0.4 for nitrite, 0.5 for silicate and
0.4 for phosphate.  Over that period |Z|-scores were satisfactory for all results for which Z-scores
were returned (>LOQ) with the exception of a single silicate result (Z = 2.04).
On the MI system every sample was analysed twice and relative percentage differences (RPDs)
were calculated for replicates, Equation 3. If any RPDs were >10%, that sample was either re-
analysed or flagged as questionable in the final dataset.
Equation 3;      $Replicate\ RPD = \frac{Replicate\ A - Replicate\ B\ concentration}{Average\ nutrient\ concentration} \times 100\%$





On the Dal system triplicate samples were measured for each sample. The coefficient of variation
was calculated (CV %) for each triplicate (Eq. 4). If the CV (%) was greater than 5 and there was
an obvious outlier, then it was rejected (max. 1 replicate of the 3 was rejected). As long as the CV
(%) for the two replicates was now < 5, the sample was accepted and not re-analyzed. For samples
with low concentrations (<0.5 μmol/l), the CV(%) was ignored unless there was an obvious
outlier, as a difference of 0.01 μmol/l between replicates would cause the CV(%) to be too high
for the lower concentrations. For samples with concentrations >10 μmol/l, outliers were
removed if the CV (%) was greater than 3. Any samples that did not pass this CV (%) test after
rejecting an outlier were rejected and re-analysed during the following run using a duplicate
sample.
Equation 4;     $$CV\% = \frac{\text{Standard deviation of replicates}}{\text{Average of replicates}} \times 100\%$$

The limit of detection (LOD) and limit of quantification (LOQ) for both instruments were
calculated as 3*standard deviation (LOD) and 10*standard deviation (LOQ) from 10 replicates of
low nutrient seawater solution, and are given in Table 4 below. Concentrations that fall between
the LOD and LOQ value are reported as <LOQ, while concentrations lower than the detection limit
are reported as <LOD.
Both systems analysed a drift sample after every 4 samples during the run to correct for
instrumental drift. The drift was made from secondary stock and artificial seawater (see
concentrations in Table 2).
System Suitability Standards (SSS) were made alongside the daily standards by the MI group
using secondary stock standards and artificial seawater.  They were analysed as an internal
standard every 4 samples to ensure drift correction is accurate and to identify any problems
during the course of a run. All SSS were checked in post processing: any that fell > ±10% of the
SSS value were marked as failed QC. Samples on either side of a failed SSS had to be re-analysed
or were flagged as questionable in the final dataset. The Dal group ran their drift sample as an
unknown to act as a system suitability standard; this was also done every four samples, but
between drift samples.  Although the drift check was monitored throughout the run, there was
no post-processing rejection based on a SSS on the Dal system, instead samples were individually
rejected based on poor replicates or an entire run was rejected if the CRMs did not pass.

**2.4 Comparison of data**
To compare the final nutrient concentrations between the two instruments the sample relative
percentage difference (RPD) was also calculated between the MI and Dal nutrient
concentrations;
Equation 5.     $$Sample\ RPD = \frac{Average\ MI\ concentration - Average\ Dal\ concentration}{Average\ nutrient\ (MI+Dal)\ concentration} \times 100\%$$
While nitrite was analysed on both instruments, there were issues with nitrite contamination in
both systems, potentially due to the Milli-Q water. While all frozen samples were re-analysed at
the MI after the survey with high quality data, a comparison of the nitrite methods and profiles
will not be carried out in this study.




## 3. Results

3.1 Comparison of instrument calibrations

Optimal calibration ranges for nutrient analysis depends on the concentrations being measured, but will also be specific to individual instruments and laboratory methods. The Dal system typically operates with a higher calibration range for all nutrients relative to the MI system, attributed to their higher volume of reagents relative to sample going through the analyser (Table 2). The MI instrument was initially established as a laboratory instrument, with high sample volumes relative to reagents to allow for precise measurements of low nutrient concentrations. The normal calibration ranges for TOxN and silicate was 0-15 µmol/l and 0-1.5 µmol/l for nitrite and phosphate. In normal laboratory use, any sample concentration outside this range is diluted into the calibration range using artificial seawater, with both sample and diluent volumes weighed accurately, and re-analysed. Because an analytical balance could not be used at sea, tests were carried out to determine the maximum range of the calibration standards, without compromising the low concentration nutrients. Phosphate and nitrite maintained linear calibrations to over 2.2 µmol/l without any changes to the methods, and therefore covered the full range of expected concentrations for the North Atlantic. With a small increase in reagent concentrations relative to sample volume, the calibration range increased to 0-30 µmol/l for TOxN and 0-60 µmol/l for silicate. Despite these changes the MI system typically had a greater sample volume relative to reagents for TOxN and silicate compared with the Dal system.

Early in the survey a negative bias was observed in the MI QC charts for the higher TOxN CRM (Batch BW, 24.6 µmol/l), while a comparison of the MI and Dal datasets also identified a negative bias in the MI TOxN data relative to the Dal data for samples from deeper in the water column (at concentrations > 15µmol/l). The reason for the bias was unclear. The TOxN calibration range on the MI system was increased from 0 – 30 µmol/l to 0 – 50 µmol/l to match the Dal system to determine if that had any effect on the TOxN QC comparison. This in fact reduced the negative bias in the BW CRM, without affecting the CD CRM (Fig. 2). Calibration standards up to 60µmol/l were analysed with all previous runs on the MI system to allow for the higher silicate range, which allowed the earlier runs to be recalculated to include standards up to 50 µmol/l.

Despite the 0-30 µmol/l range yielding the most accurate CRM values on the MI system before and after the survey (which would be expected since the MI instrument is configured for running lower nutrient concentrations), the 0-50 µmol/l range improved the higher concentration CRMs throughout the A02 survey. It is unclear why the method performed differently on the survey; a possibility is that it was due to a slight change in the light path of the photometer from ship vibrations which were more evident at the location of this containerised laboratory. However, the extra QC performed throughout the survey (two CRM batches of high and low concentration, extra calibration standards, internal SSS, a comparison with Dal and WOCE data) ensured the final results are of high quality.

A calibration test was carried out in the MI laboratory following the survey, where two rounds of 14 Quasimeme Proficiency test materials with a wide range in nutrient concentrations, were analysed with three batches of KANSO CRMs. The full suite of calibration standards (Table 2) were analysed during the run, while in the post-processing, results were exported selecting different standards and calibration coefficients (either first or second order calibration). This test was repeated a number of times and results illustrate that the range of calibration standards used can indeed have a significant effect on the final value, particularly in the low nutrient concentrations (Table 5). While nitrite and phosphate were also analysed in this experiment, the range used on



the survey were not extended beyond 2.2 µmol/l and adjusting the lower calibration standards
had minimal effect on the final concentrations. Therefore, only TOxN and silicate are discussed in
this section.
For silicate, the use of different calibration standards had marginal effect in the mid and high
sample concentrations, where almost all │Z│scores were < 1 (all <4% bias). The only samples
that illustrated a significant difference were those with concentrations < 2 µmol/l, where │Z│
scores increased to 2 if the higher calibration standards were included. For example, in the QNU
sample (Table 5), when using standards only up to 10 µmol/l, the measured value had a
difference of 7% relative to the assigned value, which was increased to 21% if standards up to 60
µmol/l were included. There was more variation in the TOxN results depending on which
standards were selected, but again it is clear that including the highest standards to 50 µmol/l
results in a larger bias in the accuracy of low concentration TOxN samples. In the QNU 307 sample,
the measured value was exactly the same as the assigned value (0% difference) if only standards
up to 10 µmol/l were included, while the difference increased to ±19% if standards up to 50
µmol/l were included. This is likely specific to the MI Skalar system as it will depend on how the
instrument can measure both high and low concentrations of nutrients and the true linearity of
the calibration standards.
Following this calibration experiment and the finding that the lowest TOxN and Silicate
concentrations showed less bias when using a smaller calibration range, the MI GOSHIP A02 data
was recalculated, where TOxN and silicate concentrations below 5µmol/l were recalculated to
only include standards up to 10 µmol/l (Table 2).
Another important finding from this experiment concerns the differences that can arise by
selection of first or second order calibration curves. GO-SHIP guidelines currently state that either
first or second order calibrations can be used but that forcing a linear fit to non-linear calibration
data can lead to offsets of 3%. It is clear that TOxN can change very significantly in the higher
concentration range, where the difference between the 1st and 2nd order calibration is close to
10% of the certified value of the CJ CRM and 8% of the BW CRM. This firmly supports the
recommendations of Hydes et al. (2010) concerning the importance of understanding and
evaluating the best fit for an individual CFA system.

**3.2 Comparison of QC between systems**
Both systems used the same Quasimeme z-score criteria for assessing the CRMs during the
survey, and all CRMs had │Z│-scores within 2, see QC charts in Fig. 3. The Dal system primarily
used the KANSO CD CRM, but ran a small number of BW CRMs for comparison towards the end of
the survey. Despite passing the assigned CRM assessment criteria, there was a negative bias in
the MI TOxN CD CRM (average difference -4%) while Dal measurements were closer to the
certified value. Silicate CD measurements were similar between the two systems, and while
phosphate CD measurements were closer to the certified value on the MI system, the Dal
phosphate QC improved later in the survey following the inclusion of more standards in the lower
range. The CV% for the CRMs (calculated as per Eq. 3) were typically below 5% for all nutrients,
Table 6.








### 3.3 Vertical profiles

Overall there was good agreement between vertical profiles of nutrients between the two systems, see Fig. 4 and Supplementary Material, giving confidence in both the overall dataset and individual methods from each group.

Looking at individual profiles of silicate, 90% of all samples compared have relative percentage differences (RPDs) < 10%, with 70% of samples with RPD < 5%. The largest differences between the two systems are in the top 400m (Fig. 5), which typically had < 6μmol/l TOxN, 3μmol/l silicate and 0.4 μmol/l phosphate, where 8% of all the samples have RPD's between 11 – 117%, with the highest RPD's in the stations with lowest silicate values.

TOxN vertical profiles also compare well with 97% of all TOxN compared with a RPD < 10%, with 77% of all RPDs < 5%. Virtually all TOxN samples with RPD > 10% are within the top 200m where TOxN values are low (Fig. 5).

Despite slightly less comparability in phosphate between the two systems; 79% of all samples had RPDs < 10%, with 38% of samples with RPD < 5%.  Almost half of the samples with RPDs > 10% were in the top 400m (Fig. 5). The remaining samples with higher differences deeper in the water column were analysed in the first three stations of the Dal system when they were encountering problems with their phosphate channel. QC of Dal phosphate improved after the they increased the number of phosphate standards in the lower concentration range, where the CV% of the CD CRM decreased from 15% in the first three runs to 7.5% in subsequent runs. This subsequently improved the comparison between the two systems.

### 3.4 Comparison with WOCE data and methods

Nutrient analysis on the WOCE A02 survey in 1997 was also carried out on a Skalar Continuous Flow Auto-Analyser (SA 4000) for photometric determination of nitrate, nitrite, phosphate and silicate. Analytical methods were similar to the MI and Dal systems, with nutrients measured at the same wavelengths, while calibrated flasks and pipettes were also used for the daily calibration standards. There were no CRMs available for the 1997 cruise, instead the internal consistency of the nutrient measurements between cruises were assessed by comparison of quality controlled dissolved inorganic carbon (DIC) data, where any inaccuracies in the nutrient measurements would show up as offsets or slope changes in the DIC-nutrient plots derived from various cruises. The estimated accuracy on the WOCE survey was 0.02 μmol/l for nitrite, 0.1 μmol/l for nitrate, 0.05 μmol/l for phosphate and 0.5 μmol/l for silicate. There was no information provided in the cruise report, and no articles published (that we know of) which states the calibration range used on this survey. The vertical profiles of nutrient data compared quite well with the 2017 data (Fig. 4). Not every station on the 2017 survey could be compared with the 1997 survey due to differences in some station positions, which coincided in bottom depth differences of over 500m between the two surveys.




## 4. Discussion

The comparison of the MI and Dal datasets from the A02 survey highlights the importance and effectiveness of following standard protocols for the sampling and analysis of nutrients at sea. Both groups followed the GO-SHIP manual for the sampling and determination of nutrients in seawater, while also incorporating their existing laboratory QC methods that were specifically adapted to their instruments.

One of the key findings in this study is the need for using two (or more) reference materials for nutrient analysis that covers the range of the expected nutrients for the survey. Hydes et al. (2010) also recommend the use of CRMs to improve the comparability of the global ocean nutrient data set, and that a minimum of three reference material solutions (low, mid and top range) should be used at regular intervals during a cruise to detect non-linearity. If only the CD CRM was used by both groups on the A02 survey, the negative bias in the MI TOxN at high concentrations would not have been identified. Without confirmation from the higher concentration CRM (Batch BW), it would not have been clear whether there was a negative bias in the MI data or a positive bias in the Dal data since both were producing similar values for the lower (CD) CRM. Although following all GO-SHIP guidelines and carrying out sufficient testing prior to the survey, there was an unexplainable change in QC in the at-sea analysis on the MI system. This highlights the necessity of including additional QC measures (e.g. high number of standards and CRMs) to allow for adjustments to the method while carrying out analysis at sea.

Results from 59 laboratories during the 2015 IOCCP-JAMSTEC inter-comparison exercise (2015 I/C exercise) indicate that non-linearity of the calibration curves for nutrient analysis is one of the significant sources of reduced comparability of nutrients data, and they also suggest that a set of reference materials should be used during analysis to cover the full range of nutrients expected (Aoyama et al., 2016). This is supported in our A02 inter-comparison, where the use of a high and low concentration CRM was able to identify analytical biases that were subsequently corrected through adjustments in the internal calibrations.

Hydes et al. (2010) suggest that the use of CRMs along with best practice in using analysis equipment and internal standardisation, should make it "commonly possible to achieve comparability of nutrient analysis to a level better than 1%". However, the ability to compare datasets to 1% will depend on the level of accuracy each laboratory can achieve. When comparing the MI and Dal nutrient data, the sample RPDs of < 1% accounted for less than 24% of samples. Below 400m, the comparison of sample concentrations results in average absolute RPD of 3.2% TOxN, 2.7% silicate and 3.7% phosphate (if the first 3 stations on the Dal system were removed).

In the 2015 I/C exercise, Aoyama et al. (2016) reported CV % of 1% TOxN, 2% silicate and 6% phosphate for the reference material batch BU (similar to Batch CD used on the A02 survey), and 2% for all nutrients for batch CA (similar to Batch BW). These CV% are lower than those produced by the MI and Dal groups on the A02 survey which were 4% for TOxN and phosphate and 5% for silicate by the MI group and 3% for TOxN, 4% silicate and 9% for phosphate by the Dal group. The CV% for the participating laboratories of the 2015 I/C exercise were calculated from measurements carried out in shore-based laboratories, a much more stable environment than at carrying out analysis at sea. Higher error in measurements of reference materials analysed at sea could be due to the use of pipettes (as opposed to balances) for daily calibration standards, different Milli-Q water supply, pre-weighed chemicals for reagents, different analysts and a moving platform with vibrations that could influence the light path of the instrument. The CV% of the KANSO CRMs (Batch CD) analysed in the MI laboratory (on shore) since the A02 survey was reduced to 3% for all nutrients (n=24).





In another inter-comparison study carried out in 2005 and 2006 (Sahlsten and Håkansson, 2006),
five different laboratories from the monitoring institutes of Denmark, Norway and Sweden,
compared nutrient concentrations from identical sets of natural seawater sub-samples (as
opposed to prepared reference materials) that were analysed ashore in individual laboratories;
results for the deep water samples indicated precision generally less than 5% CV between
laboratories. This study indicated that the variation between laboratories could be explained by
improper storage of the nutrient samples between sampling and analysis. Our results, however
suggests that this level of comparability could instead be due to systematic differences between
instruments and individual internal calibrations. Tanhua et al. (2009b) and Tanhua et al. (2009a)
carried out cross over analysis as a secondary QC on the nutrient data in the Atlantic (CARINA),
where an offset and standard deviation were calculated for nutrients at depths >1500m. They
found nitrate data showed the largest consistency with RMSE of 2.9%, with a RMSE of 4.2% for
phosphate and 7% for silicate, and suggested the larger differences in the data were likely due to
analytical difficulties.
With availability of a range of CRMs for nutrients in seawater, there remains a need for
acceptability criteria for oceanic nutrient measurements to meet GO-SHIP objectives. Such
criteria exist for other biogeochemical parameters, for example, for dissolved inorganic carbon
(DIC) and total alkalinity (TA) in the open ocean, a level of accuracy of ±2 µmol/kg for reference
materials, (~ 0.1%), is recommended in order to assess annual or even decadal changes in the
marine carbonate system (Dickson, 2010; ICES, 2014). In coastal waters, the level of accuracy
required would be less since the range of carbonate parameters observed would be much wider
than those in the open ocean. If criteria for nutrient measurements were set, laboratories could
flag reported data where these were not achieved. The metadata supplied with published datasets
should include all of the related QC information, including calibration ranges, batches of CRMs
used, CRM assessment criteria, accuracy of CRMs achieved, sample storage prior to analysis, etc.
The largest differences between the MI and Dal nutrient concentrations were in the surface
waters, where low levels of nutrients were observed due to primary production. Reduced
comparability between the participating laboratories of the 2015 I/C exercise (Aoyama et al.,
2016) was also observed in the low nutrient reference materials, which yielded CV% of up to
60%. Larger differences in low nutrient waters would be expected since any error in calibration
standards, instrument baselines and detection limits would strongly impact concentrations close
to the limit of detection. The MI instrument runs ASW as a baseline wash, while the Dal instrument
runs Milli-Q water; while this could result in differences in low nutrient samples, it is unlikely to
be the issue here since both groups were using the same Milli-Q water supply to make reagents
and wash and the sodium chloride used for the ASW on the MI system was tested ahead of the
survey to ensure no contamination in the batches used. The large differences in the low nutrient
concentrations is instead likely due to the sample:reagent ratio of each system, where the
instruments have different capability of measuring low nutrient concentrations.
It would appear from the vertical profiles that the low nutrient surface waters (<400m) would
have little relevance when looking at the overall vertical distribution of nutrients across the North
Atlantic. And while its significance would be minimal in comparing nutrient concentrations in
intermediate and deep water masses, inaccurate nutrient concentrations in the euphotic zone
would lead to large discrepancies in primary production estimates and near-surface N:P ratios.
In the entire GO-SHIP A02 survey, 32% of all samples are within the top 400m of the water
column, and therefore represent a large proportion of the entire dataset. Clearly, achieving high
accuracy measurements across the large concentration ranges that are encountered from surface
to deep waters remains an analytical challenge. It is difficult to compare upper water column





nutrients in cross-over analysis based on different cruises in the same area due to more variability
on different time-scales (Tanhua et al., 2009a; Tanhua et al., 2009b). This inter-comparison study
therefore addresses a key issue in comparability of nutrient data in low nutrient surface waters.
There are currently no KANSO CRMs that have concentrations close to detection limits to quantify
bias in low nutrient surface waters, which perhaps should be considered for the future.

The results of this inter-comparison strongly support the suggestions in Hydes et al. (2010) that
individual laboratories or groups must carry out extensive internal testing on their own
instruments to understand the full capability of their instruments and ensure their laboratory
methods achieve the highest level of accuracy for the samples being measured. Results also
highlighted the value of carrying out a between-laboratory testing exercise, which in this case,
helped both groups to identify quality assurance issues in their internal procedures which would
otherwise not have been evident. All laboratory groups should ensure they incorporate additional
QC into their methods, including extra calibration standards, extra reference materials and
internal standards, to allow for post-correction of data if some unforeseen changes to their
instrument occurs while at sea.

**5. Data Availability**

The GO-SHIP A02 nutrient dataset (analysed on the Marine Institute Skalar nutrient analyser) is
currently available at the National Oceanographic Data Centre of Ireland;
http://data.marine.ie/publication/dataset/ce49bc4c-91cc-41b9-a07f-d4e36b18b26f.html.
http://dx.doi.org/10.20393/CE49BC4C-91CC-41B9-A07F-D4E36B18B26F

**6. Conclusions and Recommendations**

For data to be of use to the scientific community, oceanographic data collected by different groups
must be comparable in order to assess true changes in the marine environment. The presence of
biases or imprecision in the measurement of nutrients in seawater reduce our ability to
understand spatial and temporal trends in nutrient concentrations in the ocean. The comparison
of two nutrient datasets from the 2017 A02 survey illustrated high quality control in the
analytical methods and high comparability between datasets, highlighting the effectiveness of
following standard protocols and using CRMs while at sea. The cross-comparison of laboratory
methods, quality control and instrument configurations also allowed the MI and Dal groups to
scrutinize their laboratory procedures in order to identify reasons for analytical bias while
carrying out nutrient analysis at sea. Following this study, a number of recommendations are
suggested which enhance those in the GO-SHIP manual (Hydes et al., 2010) for improved quality
of global nutrient datasets;

- Multiple (At least two) CRMs should be used that cover the range of the expected concentrations on the survey to assess linearity and identify any analytical bias at different concentrations.
- Agreed CRM acceptance criteria for ocean observation nutrient measurement would aid in improving data quality and support flagging of reported data that doesn't meet these criteria
- Extensive testing must be carried out ahead of a survey to understand individual instrument capabilities and extra QC should be included to allow for changes to the methods due to unforeseen changes while carrying out analysis at sea.
- Metadata should include all information related to QC so to increase comparability and traceability between different nutrient datasets.



**Acknowledgements**

Financial support for the survey was provided by the Marine Institute, under the Irish Government's Marine Research Programme 2014-2020, with support from the AtlantOS project - funded by the European Union's Horizon 2020 research and innovation programme under grant agreement No. 633211 and the Canada Excellence Research Chair in Ocean Science and Technology. The survey represents an initial activity of the newly-formed Ocean Frontier Institute of which the Irish Marine Institute and GEOMAR are partners. We would like to thank the crew and scientists on board the RV Celtic Explorer on the A02 survey, and the various support teams at the Marine Institute.

**Competing interests**

The authors declare that they have no conflict of interest.

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



Table 1. A comparison of sampling, instrument configurations (including sample and reagent tubing sizes)
and reagent compositions for each nutrient from the Marine Institute, Ireland (MI) and Dalhousie
University, Canada (Dal) systems.

| | MI | Dal |
|---|---|---|
| **Sampling** | | |
| Sample tubes | 50ml falcon tubes | 15 ml falcon tubes |
| Primary sample analysis | Within 12 hours of sampling | Within 12 hours of sampling |
| Replicate samples | Frozen immediately to -20°C | Stored at 4°C and analysed within 36 hours if necessary |
| **Analysis** | | |
| Auto-sampler size | 300 cups | 50 cups (can be re-filled during a run) |
| Auto-sampler cup size | 10ml | 4ml |
| Baseline wash | Artificial Seawater | Milli-Q water |
| **Reagents (Chemicals g/L or ml/L)** | | |
| Artificial Seawater | 35g Sodium Chloride | 35g Sodium Chloride |
| | 0.5g Sodium hydrogen carbonate | |
| **TOxN** | | |
| Sample tubing size | 1.02 ml/min | 0.16 ml/min |
| **Colour Reagent** | 150ml Phosphoric Acid | 150 ml Phosphoric acid |
| | 10g Sulfanimide | 10 g Sulfanilamide |
| | 0.5g N-(1-Naphthyl)ethylene diamine dihydrochloride (NEDD) | 0.5 g NEDD |
| | | 6 ml Brij solution |
| Reagent tubing size | 0.42 ml/min | 0.42 ml/min |
| **Buffer Solution** (pH 8.2) | 80g Ammonium Chloride | 17.5 g Imidazole |
| | ~3ml Ammonia Solution | ~25 ml 1M Hydrochloric Acid |
| | 3ml Brij solution (surfactant) | 1 ml Brij solution |
| Reagent tubing size | 0.8 ml/min | 1.6 ml/min |
| **Cadmium column** | Skalar 5358 activated Cd column | Skalar 5347 nitrate reduction coil |
| **Copper Sulfate Solution** | | 12 g copper sulfate |
| **Nitrite** | | |
| Sample tubing size | 0.42 ml/min | 1.20 ml/min |
| **Colour Reagent** | 150ml Phosphoric Acid | 150 ml Phosphoric acid |
| | 10g Sulfanilamide | 10 g Sulfanilamide |
| | 0.5g NEDD | 0.5 g NEDD |
| | | 6 ml Brij solution |
| Reagent tubing size | 0.23 ml/min | 0.23 ml/min |
| **Wash Solution** | 3ml Brij solution | NA |
| Reagent tubing size | 1.00 ml/min | |
| **Silicate** | | |
| Sample tubing size | 1.40 ml/min | 0.42 ml/min |
| **Sulfuric Acid Solution** | 20ml Sulfuric Acid | 5 ml Sulfuric acid |
| | | 1 g Lauryl sulfate |
| Reagent tubing size | 0.23 ml/min | 0.42 ml/min |
| **Ammonium heptamolybdate** | 20g Ammonium heptamolybdate | 10 g Ammonium heptamolybdate |
| Reagent tubing size | 0.42 ml/min | 0.42 ml/min |
| **Oxalic Acid** | 44g Oxalic Acid | 44 g Oxalic acid |



| | | |
|---|---|---|
| Reagent tubing size | 0.42 ml/min | 0.42 ml/min |
| **L(+) Ascorbic Acid** | 40g Ascorbic Acid | 40 g Ascorbic acid |
| Reagent tubing size | 0.32 ml/min | 0.32 ml/min |
| **Phosphate** | | |
| Sample tubing | 1.40 ml/min | 1.60 ml/min |
| **Ammonium heptamolybdate** | 0.23g Potassium antimony (III) | 0.23 g Potassium antimony (III) oxide |
| | 70ml Sulfuric Acid | 70 ml Sulfuric acid |
| | 6g Ammonium heptamolybdate | 6 g Ammonium heptamolybdate |
| | 2ml FFD6 (Surfactant) | 5 ml FFD6 |
| Reagent tubing size | 0.42 ml/min | 0.32 ml/min |
| **L(+) Ascorbic Acid** | 11g Ascorbic Acid | 11 g Ascorbic acid |
| | 60ml Acetone | 60 ml Acetone |
| | 2ml FFD6 | 5 ml FFD6 |
| Reagent tubing size | 0.42 ml/min | 0.32 ml/min |



























Table 2. Concentrations of daily calibration standards in µmol/l on the MI and Dal systems. Standard 1 is
the blank made of artificial seawater (sal 35). Standards 2-4 with the * on the Dal system were added to
their standards only on the last 4 days of analysis following discussions with the MI group. SSS are the
system suitability standards that were analysed during a run as internal quality standards.

| STD # | MI | | | | Dal | | | |
|---|---|---|---|---|---|---|---|---|
| | TOxN µmol/l | Silicate µmol/l | PO4 µmol/l | NO2 µmol/l | TOxN µmol/l | Silicate µmol/l | PO4 µmol/l | NO2 µmol/l |
| 1 | 0 | 0 | 0 | 0 | 0 | 0 | 0 | 0 |
| 2 | 0.26 | 0.26 | 0.05 | 0.05 | 1.25 * | 1.25 * | 0.1 * | 0.15 * |
| 3 | 0.5 | 0.5 | 0.15 | 0.15 | 2.5 * | 2.5 * | 0.2 * | 0.3 * |
| 4 | 2.5 | 2.5 | 0.25 | 0.25 | 5 * | 5 * | 0.4 * | 0.6 |
| 5 | 5 | 5 | 0.5 | 0.5 | 10 | 10 | 0.8 | 1.2 |
| 6 | 10 | 10 | 1 | 1 | 20 | 20 | 1.6 | 1.8 |
| 7 | 15 | 15 | 1.5 | 1.5 | 30 | 30 | 2.4 | 2.4 |
| 8 | 22.5 | 22.5 | 2.25 | 2.25 | 40 | 40 | 3.2 | 3.0 |
| 9 | 30 | 30 | | | 50 | 50 | 4.0 | |
| 10 | 40 | 40 | | | | | | |
| 11 | 50 | 50 | | | | | | |
| 12 | | 60 | | | | | | |
| SSS | 10 | 10 | 1 | 1 | 40 | 40 | 3.2 | 2.4 |
| Drift | 10 | 10 | 1 | 1 | 40 | 40 | 3.2 | 2.4 |
























Table 3. Certified values for the two batches of KANSO CRMs used on the survey.

| Certified values µmol/l | | |
|---|---|---|
|  | CD | BW |
| Nitrate | 5.498 | 24.59 |
| Nitrite | 0.018 | 0.067 |
| Silicate | 13.93 | 60.01 |
| Phosphate | 0.446 | 1.541 |






























Table 4. The limit of detection (LOD) and limit of quantification (LOQ) in μmol/l, for both instruments.

|  | MI | | | | Dal | | | |
|---|---|---|---|---|---|---|---|---|
|  | TOxN | Nitrite | Silicate | Phosphate | TOxN | Nitrite | Silicate | Phosphate |
| LOD | 0.02 | 0.01 | 0.03 | 0.01 | 0.14 | 0.02 | 0.13 | 0.04 |
| LOQ | 0.26 | 0.04 | 0.38 | 0.16 | 0.48 | 0.07 | 0.43 | 0.13 |































Table 5. Results from a laboratory experiment testing the effect of using different calibration ranges,
where STD in the first column of the table indicates the top standard included in the calibration. The
second column (Order) indicates whether the first or second order calibration coefficient was used in the
calibration. The samples are either Quasimeme test materials (QNU) or KANSO CRMs; MV is the measured
value; AV is the assigned (or certified value); TE is the total error used for calculating the z-score; Z is the
calculated z-score as per Eq. 1 and RPD is the relative % difference (MV-AV/AV*100%).

| | | | TOxN | | | | | Silicate | | | | |
|---|---|---|---|---|---|---|---|---|---|---|---|---|
| STD | Order | Sample | MV | AV | TE | Z | RPD | MV | AV | TE | Z | RPD |
| 10 | 1st | QNU 304 EW | -0.04 | 0.07 | 0.03 | <LOD | | 1.97 | 2.17 | 0.18 | -1.1 | -9 |
| 22 | 1st | QNU 304 EW | -0.09 | 0.07 | 0.03 | <LOD | | 1.97 | 2.17 | 0.18 | -1.1 | -9 |
| 30 | 1st | QNU 304 EW | -0.16 | 0.07 | 0.03 | <LOD | | 1.94 | 2.17 | 0.18 | -1.3 | -11 |
| 50 | 1st | QNU 304 EW | -0.77 | 0.07 | 0.03 | <LOD | | 1.96 | 2.17 | 0.18 | -1.2 | -10 |
| 50 | 2nd | QNU 304 EW | 0.10 | 0.07 | 0.03 | <LOQ | | 1.81 | 2.17 | 0.18 | -2.0 | -17 |
| 60 | 1st | QNU 304 EW | Failed Calibration | | | | | 1.95 | 2.17 | 0.18 | -1.2 | -10 |
| 60 | 2nd | QNU 304 EW | 0.43 | 0.07 | 0.03 | 11.6 | 552 | 1.97 | 2.17 | 0.18 | -1.1 | -9 |
| 10 | 1st | QNU 307 SW | 2.16 | 2.16 | 0.16 | 0.0 | 0 | 1.91 | 2.00 | 0.17 | -0.5 | -4 |
| 22 | 1st | QNU 307 SW | 2.15 | 2.16 | 0.16 | -0.1 | -1 | 1.91 | 2.00 | 0.17 | -0.5 | -5 |
| 30 | 1st | QNU 307 SW | 2.15 | 2.16 | 0.16 | -0.1 | -1 | 1.90 | 2.00 | 0.17 | -0.6 | -5 |
| 30 | 2nd | QNU 307 SW | 2.15 | 2.16 | 0.16 | -0.1 | -1 | 1.90 | 2.00 | 0.17 | -0.6 | -5 |
| 50 | 1st | QNU 307 SW | 1.75 | 2.16 | 0.16 | -2.6 | -19 | 1.82 | 2.00 | 0.17 | -1.0 | -9 |
| 50 | 2nd | QNU 307 SW | 2.18 | 2.16 | 0.16 | 0.1 | 1 | 1.91 | 2.00 | 0.17 | -0.5 | -4 |
| 60 | 1st | QNU 307 SW | Failed Calibration | | | | | 1.72 | 2.00 | 0.17 | -1.6 | -14 |
| 60 | 2nd | QNU 307 SW | 2.22 | 2.16 | 0.16 | 0.4 | 3 | 1.92 | 2.00 | 0.17 | -0.4 | -4 |
| 10 | 1st | QNU 300 SW | 2.92 | 2.75 | 0.19 | 0.9 | 6 | 1.46 | 1.57 | 0.15 | -0.8 | -7 |
| 22 | 1st | QNU 300 SW | 2.91 | 2.75 | 0.19 | 0.8 | 6 | 1.45 | 1.57 | 0.15 | -0.8 | -8 |
| 30 | 1st | QNU 300 SW | 2.91 | 2.75 | 0.19 | 0.8 | 6 | 1.43 | 1.57 | 0.15 | -0.9 | -9 |
| 50 | 1st | QNU 300 SW | 2.57 | 2.75 | 0.19 | -0.9 | -7 | 1.35 | 1.57 | 0.15 | -1.5 | -14 |
| 50 | 2nd | QNU 300 SW | 2.87 | 2.75 | 0.19 | 0.6 | 4 | 1.46 | 1.57 | 0.15 | -0.8 | -7 |
| 60 | 1st | QNU 300 SW | Failed Calibration | | | | | 1.25 | 1.57 | 0.15 | -2.2 | -21 |
| 60 | 2nd | QNU 300 SW | 2.89 | 2.75 | 0.19 | 0.7 | 5 | 1.47 | 1.57 | 0.15 | -0.7 | -6 |
| 10 | 1st | QNU 299 SW | 6.69 | 6.75 | 0.43 | -0.2 | -1 | 5.36 | 5.36 | 0.37 | 0.0 | 0 |
| 22 | 1st | QNU 299 SW | 6.66 | 6.75 | 0.43 | -0.2 | -1 | 5.37 | 5.36 | 0.37 | 0.0 | 0 |
| 30 | 1st | QNU 299 SW | 6.50 | 6.75 | 0.43 | -0.6 | -4 | 5.34 | 5.36 | 0.37 | -0.1 | 0 |
| 50 | 1st | QNU 299 SW | 6.70 | 6.75 | 0.43 | -0.1 | -1 | 5.31 | 5.36 | 0.37 | -0.2 | -1 |
| 50 | 2nd | QNU 299 SW | 6.30 | 6.75 | 0.43 | -1.1 | -7 | 5.35 | 5.36 | 0.37 | 0.0 | 0 |
| 60 | 1st | QNU 299 SW | Failed Calibration | | | | | 5.31 | 5.36 | 0.37 | -0.1 | -1 |
| 60 | 2nd | QNU 299 SW | 6.08 | 6.75 | 0.43 | -1.5 | -10 | 5.28 | 5.36 | 0.37 | -0.2 | -2 |
| 10 | 1st | KANSO CD | 5.55 | 5.50 | 0.35 | 0.2 | 1 | | 13.93 | 0.89 | | |
| 22 | 1st | KANSO CD | 5.53 | 5.50 | 0.35 | 0.1 | 0 | 14.30 | 13.93 | 0.89 | 0.4 | 3 |
| 30 | 1st | KANSO CD | 5.53 | 5.50 | 0.35 | 0.1 | 1 | 14.34 | 13.93 | 0.89 | 0.5 | 3 |
| 50 | 1st | KANSO CD | 5.39 | 5.50 | 0.35 | -0.3 | -2 | 14.45 | 13.93 | 0.89 | 0.6 | 4 |
| 50 | 2nd | KANSO CD | 5.30 | 5.50 | 0.35 | -0.6 | -4 | 14.24 | 13.93 | 0.89 | 0.3 | 2 |
| 60 | 1st | KANSO CD | Failed Calibration | | | | | 14.51 | 13.93 | 0.89 | 0.7 | 4 |
| 60 | 2nd | KANSO CD | 5.24 | 5.50 | 0.35 | -0.7 | -5 | 14.18 | 13.93 | 0.89 | 0.3 | 2 |
| 22 | 1st | KANSO CJ | 16.08 | 16.2 | 1.00 | -0.1 | -1 | | 38.5 | 2.360 | | |
| 30 | 1st | KANSO CJ | 16.22 | 16.2 | 1.00 | 0.0 | 0 | | 38.5 | 2.360 | | |
| 50 | 1st | KANSO CJ | 17.16 | 16.2 | 1.00 | 1.0 | 6 | 39.36 | 38.5 | 2.360 | 0.4 | 2 |



| 50 | 2nd | KANSO CJ | 15.59 | 16.2 | 1.00 | -0.6 | -4 | 39.32 | 38.5 | 2.360 | 0.3 | 2 |
| 60 | 1st | KANSO CJ | Failed Calibration | | | | | 39.62 | 38.5 | 2.360 | 0.5 | 3 |
| 60 | 2nd | KANSO CJ | 15.29 | 16.2 | 1.00 | -0.9 | -6 | 39.33 | 38.5 | 2.360 | 0.4 | 2 |
| 22 | 1st | KANSO BW | | 24.59 | 1.50 | | | | 60.01 | 3.65 | | |
| 30 | 1st | KANSO BW | 24.56 | 24.59 | 1.50 | 0.0 | 0 | | 60.01 | 3.65 | | |
| 50 | 1st | KANSO BW | 26.41 | 24.59 | 1.50 | 1.2 | 7 | | 60.01 | 3.65 | | |
| 50 | 2nd | KANSO BW | 24.45 | 24.59 | 1.50 | -0.1 | -1 | 60.30 | 60.01 | 3.65 | 0.1 | 0 |
| 60 | 1st | KANSO BW | Failed Calibration | | | | | 60.05 | 60.01 | 3.65 | 0.0 | 0 |
| 60 | 2nd | KANSO BW | 24.06 | 24.59 | 1.50 | -0.4 | -2 | 60.88 | 60.01 | 3.65 | 0.2 | 1 |





























Table 6. Calculated coefficient of variation (CV%) for the KANSO CRMs analysed by the Marine Institute
(MI) and Dalhousie University (Dal), calculated as the (standard deviation/mean*100%). The KANSO
batches CD and BW were used by both groups, where n is the number of measurements.

| Nutrient | MI | | Dal | |
|---|---|---|---|---|
| | CV% | n | CV% | n |
| TOxN (CD) | 4 | 27 | 3 | 27 |
| Silicate (CD) | 5 | 27 | 4 | 27 |
| Phosphate (CD) | 4 | 27 | 9 | 27 |
| TOxN (BW) | 3 | 16 | 1 | 4 |
| Silicate (BW) | 5 | 16 | 3 | 4 |
| Phosphate (BW) | 3 | 16 | 4 | 4 |



























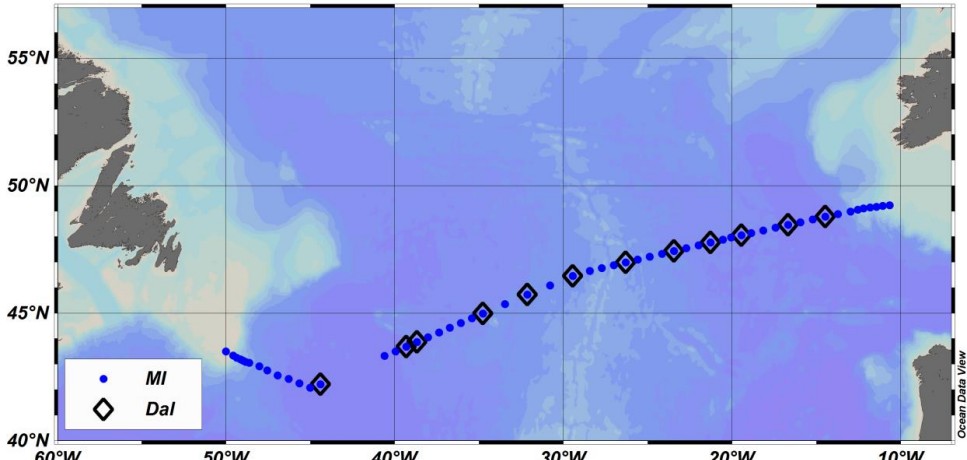


Figure 1. Station positions sampled along the GO-SHIP A02 trans-Atlantic survey completed in May 2017.
The Marine Institute (MI) group sampled and analysed nutrient samples at every station along the transect,
while the Dalhousie group (Dal) analysed nutrient samples from a selected number of sites marked with a
diamond. Both groups analysed samples over the full water column.









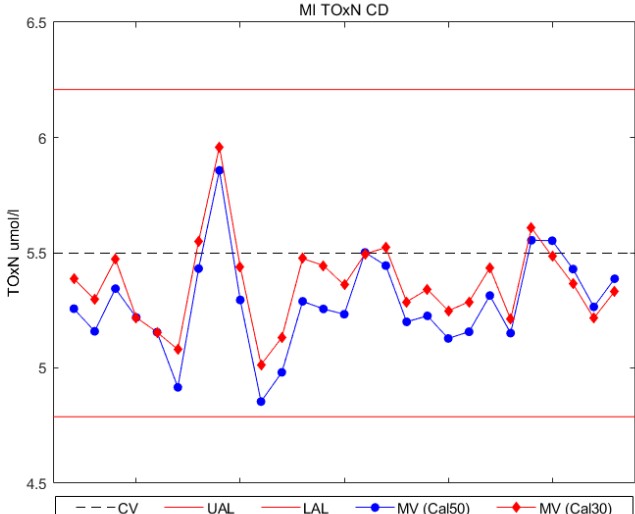

838                                                                                          Figure 2a

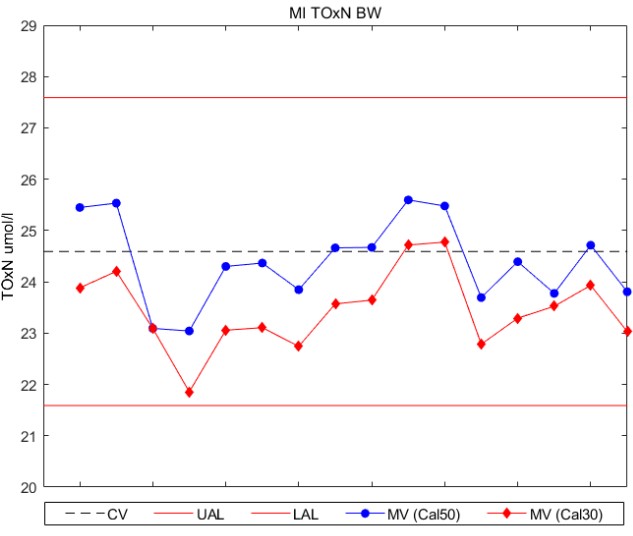

839                                                                                          Figure 2b

Figure 2 (a) and (b) Measured values for TOxN CD and BW CRMs on the MI system during the A02 survey
to illustrate the effects of using either a calibration ranges 0-50 µmol/l(Cal50) and 0-30 µmol/l (Cal30),
where CV is the certified value of each CRM and UAL and LAL, are the upper and lower action limits using
a z-score of 2 criteria. Each point represents CRM results from an individual run. Due to improved QC
using the TOxN range 0-50 µmol/l, the runs were re-calculated to include the higher standards.







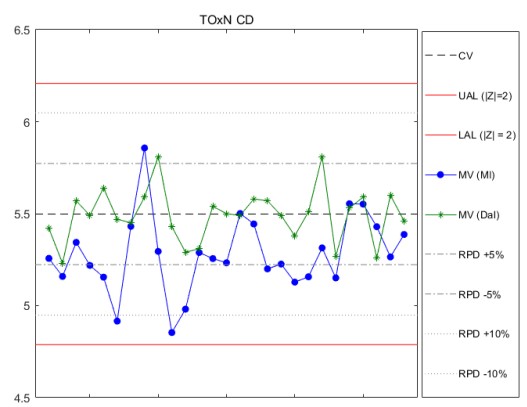

849                                                                                       Figure 3a

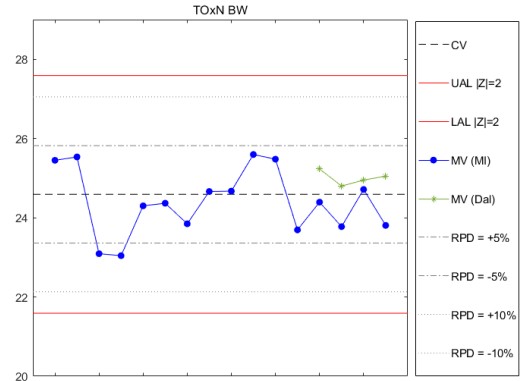

850                                                                                       Figure 3b

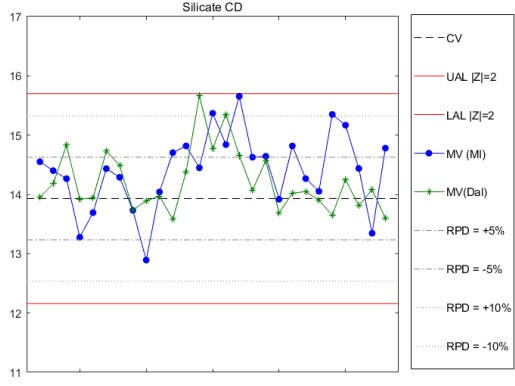

851                                                                                       Figure 3c





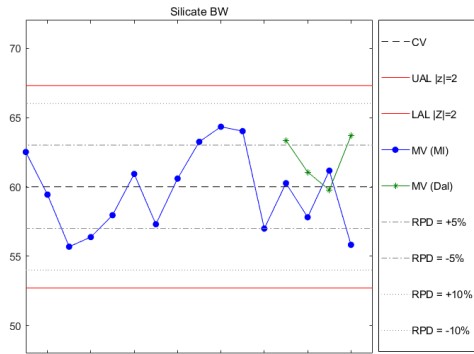

852                                                          Figure 3d

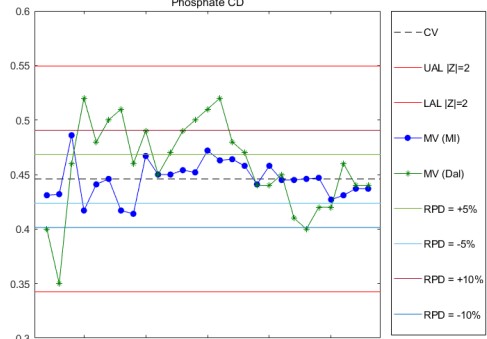

853                                                          Figure 3e

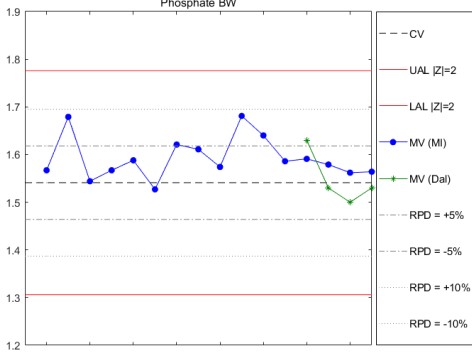

854                                                          Figure 3f

Figures 3a-3f Control charts of CRM concentrations from the MI and Dal systems. The dashed line
represents the certified value for each CRM (CV), while the red upper (UAL, upper action limit) and lower
(LAL, lower action limit) lines represent the z-score of 2 allowable limits criteria. MV (MI) and MV (Dal) are
the measured values from the MI and Dal systems, respectively. The dash-dot and dotted lines represent
the 5% and 10% relative percentage difference from the certified value. One CD CRM was run at the
beginning and end of every run on both systems, and one BW CRM was analysed at the beginning of every
run on the MI system. BW CRMs were run on only a selected number of runs of the Dal system for
comparison.





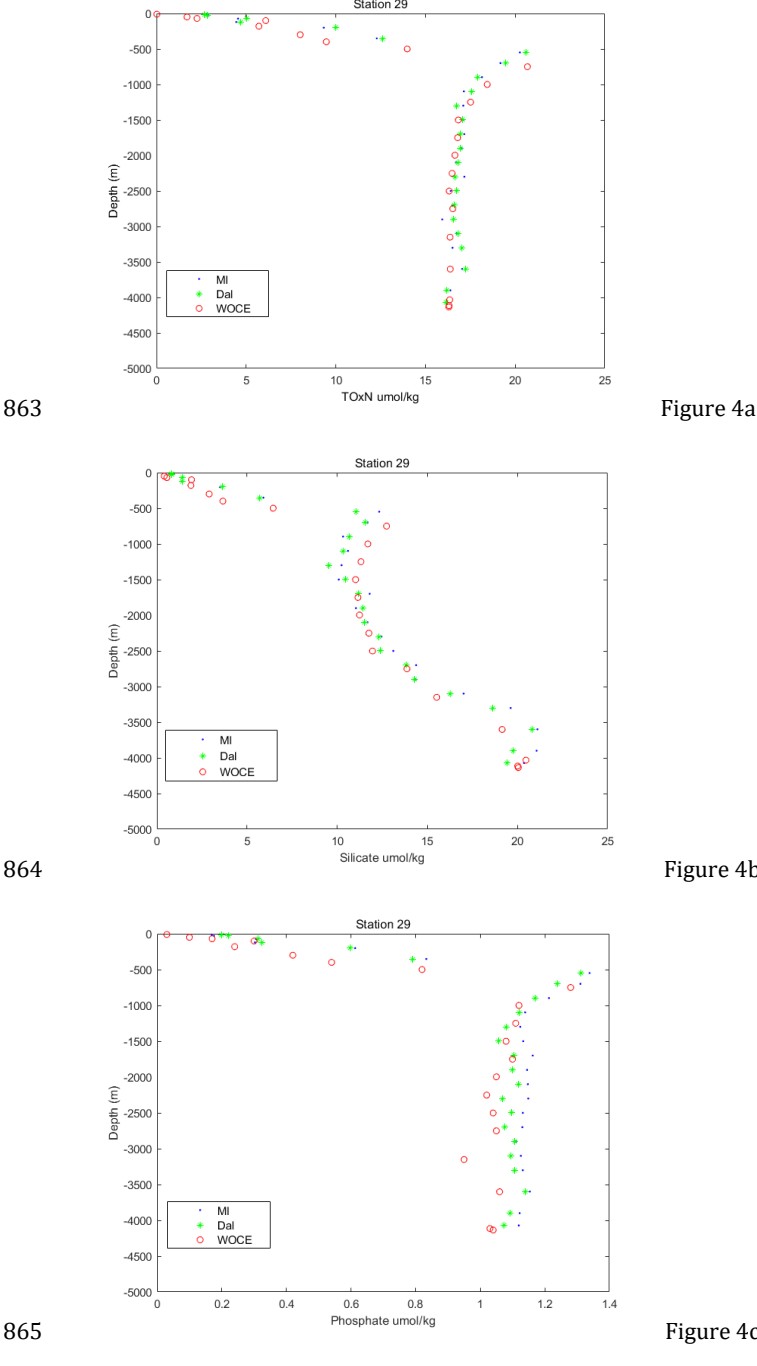

863                                                                       Figure 4a

864                                                                       Figure 4b

865                                                                       Figure 4c




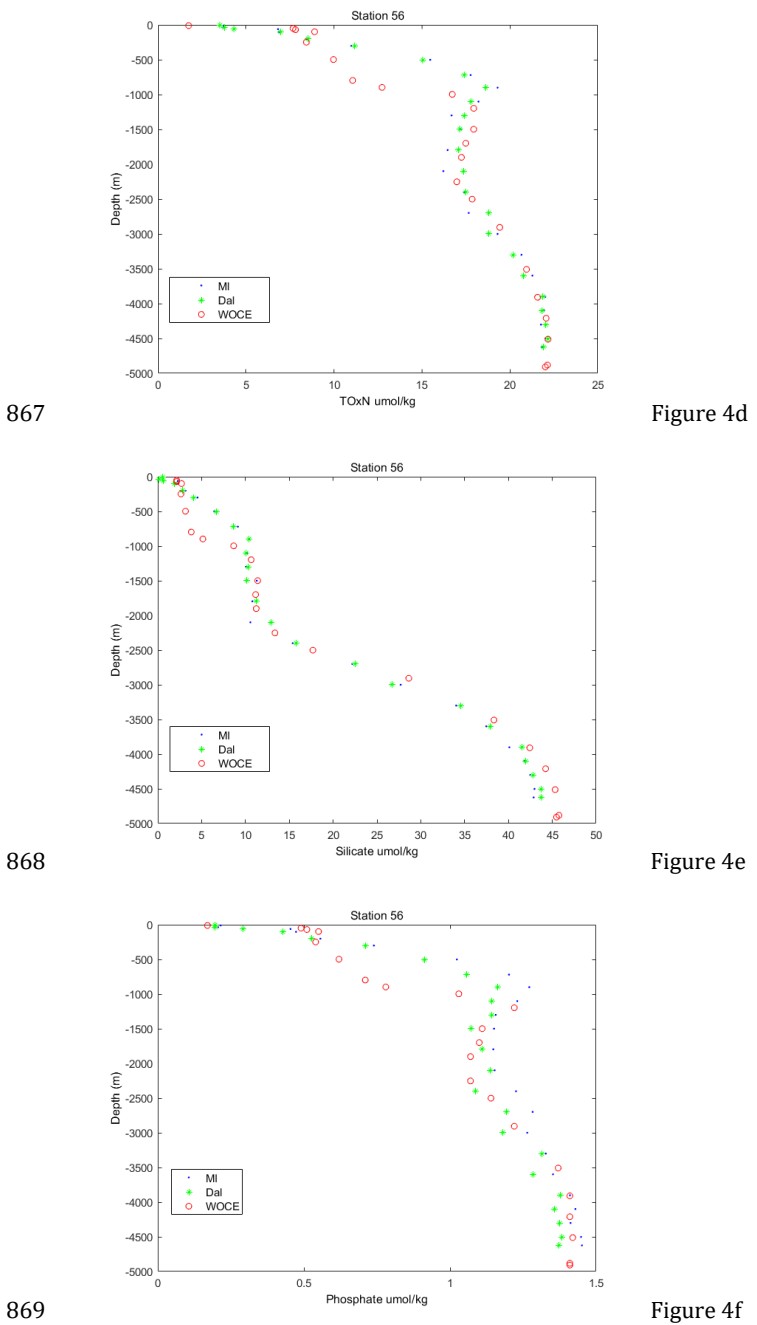

867                                                                             Figure 4d

868                                                                             Figure 4e

869                                                                             Figure 4f

Figure 4. Vertical profiles of TOxN, Silicate and Phosphate (in µmol/kg from the MI (Marine Institute), Dal
(Dalhousie University) and WOCE (World Ocean Circulation Experiment) datasets. Only station 29 and 56
are included here, all other stations compared are in the Supplementary Material.





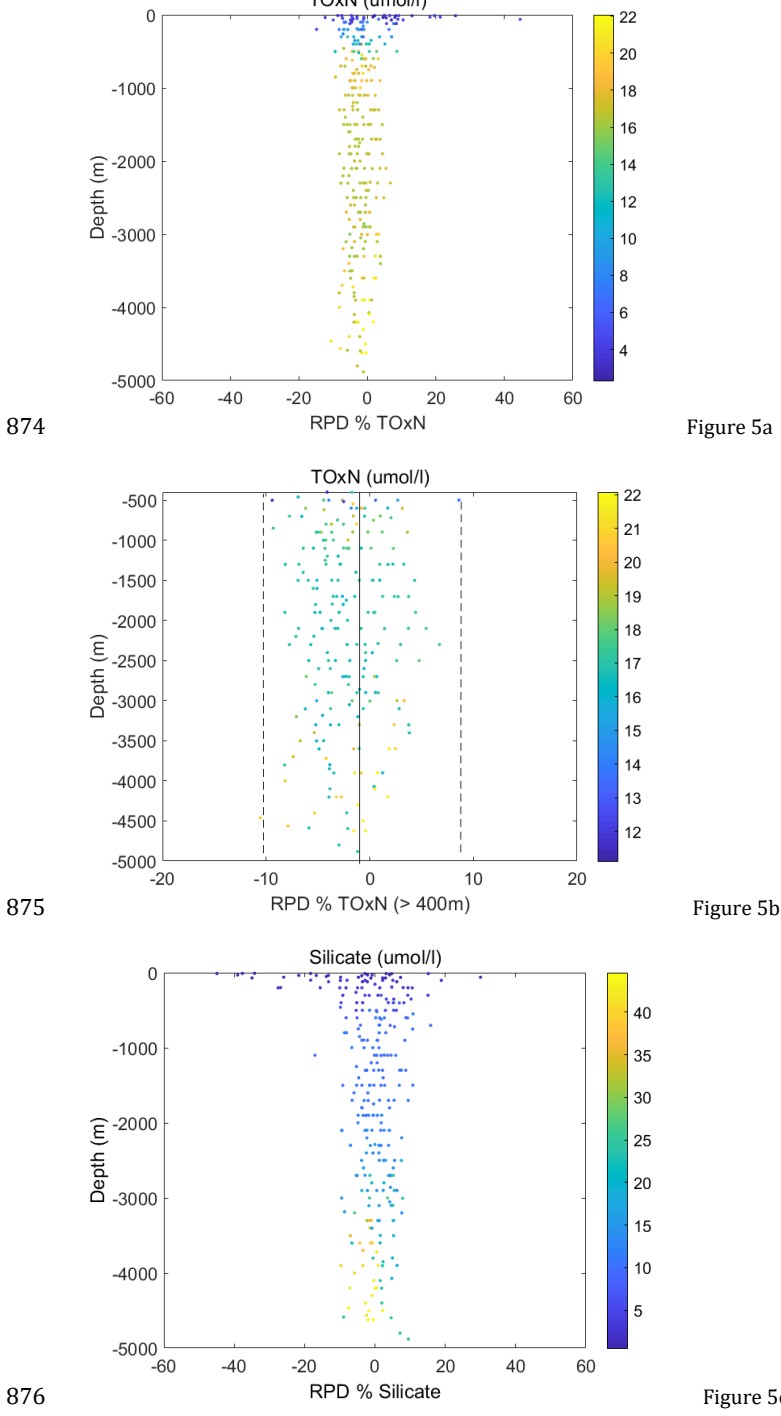

874                                                                 Figure 5a

875                                                                 Figure 5b

876                                                                 Figure 5c



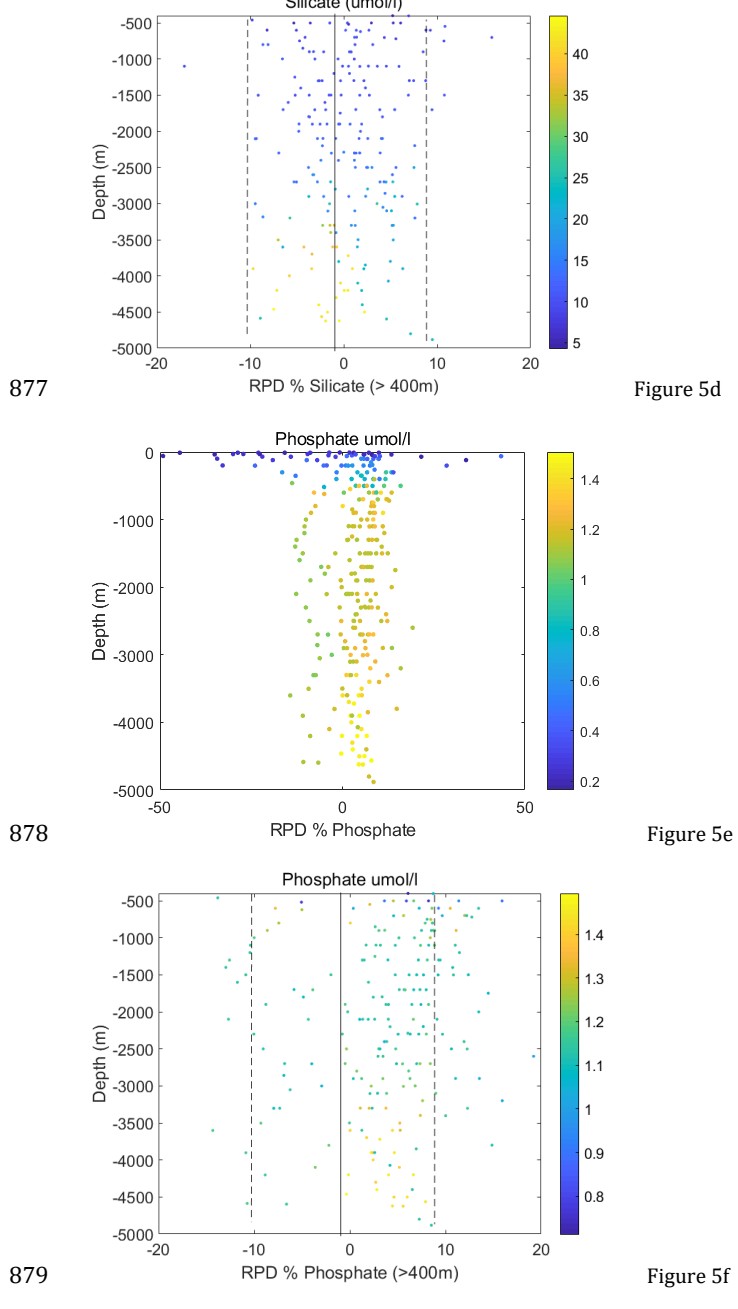

877                                                        Figure 5d

878                                                        Figure 5e

879                                                        Figure 5f


Figure 5 Relative percentage difference (RPD) calculated as (MI conc - Dal conc)/average conc * 100% for
each nutrient for the whole water column and for depths > 400m. The colour bar for each plot is the
average concentration (µmol/l) of each nutrient (i.e. the average concentration from both systems) at that
depth.