# Peer review of "Earth System Discussion Science usions"

_Earth System Science Data, 2018_

## Referee Comment (RC1) · Anonymous Referee #1 · 4 May 2018

Review of McGrath et al It is good to see this rather unusual detailed inter-comparison of sweater nutrient analyses methods at sea. I think the results will be useful for others and worth publishing although I would suggest some minor amendments. My only substantial suggestion is to make slightly more of the points touched on in the conclusions. The issue is not absolute accuracy and precision in its own right but data that is fit for purpose. In the deep ocean waters where nutrients are used to trace water flows and understand ocean scale biogeochemistry, these "fit for purpose" mean something very different to surface waters, where nutrient concentrations may be used to estimate biological production and its controls. The errors in quantifying these latter processes using the methods here, where detection limits are such that surface

water concentrations will be at or far below measurable concentrations, will be large. Improving sensitivity is then the issue. In deep waters the bias and accuracy issues are key. Additional specific minor points. Were reagents prepared freshly every day or less frequently? Line 213-5 I did not really understand how blanks were run on the Dal system with a Milli-Q wash. The blank is then the wash I assume. In addition at low concentration, how the zero is calculated becomes key, and this could be better explained. Line 270-9 I would assume drift samples were used to correct for drift, but here they seem to be being used as a quality control criteria and no drift assumed. Is this right and if so how long a batch run of samples was usually conducted? Line 276-7 I do not understand what is meant here by "but between drift samples" Section 3.1 I understand the point about why you might get slightly different results using different standard concentration ranges, but the issue is in part I would think about how you fit the best fit graph and whether you force the line through zero, so perhaps this should be clarified. The paper tends to assume the reader is very familiar with the Hydes et al paper and that may not always eb the case. Section 3.3 I wonder if the use of percentages here may not convey all the useful possible information since these can be misleading particularly at low concentrations. Discussion It seems to me that the Hydes et al suggestion that comparability of better than 1% is an extremely ambitious target. This might be realised by groups working in the laboratory and sharing common standards but that is not how ocean nutrient data is actually collected. A useful conclusion of this paper might be a more realistic evaluation of this comparability target by quoting the kind of comparability they see in different concentration ranges. Physical oceanographers can achieve agreement to 1% for salinity perhaps, but I think it is unrealistic for nutrient analyses and probably not necessary for most purposes. One last point. There is an implication that samples were stored frozen and then analysed on land, it would be interesting to know how these results compared. It may well be possible to achieve better comparability on land, but that would require that storage is effective and the work here might reveal if that is the case. A Minor details in the Phosphate method, what is FFD6?

**[ESSDD](https://doi.org/10.5194/essd-2018-47)**

Interactive
comment

---

## Referee Comment (RC2) · Anonymous Referee #2 · 31 May 2018

Review of McGrath et al. Novel nutrient comparison

The authors present data from a GO-SHIP cruise where they have the relatively unique opportunity to have two nutrient autoanalyzers on the same cruise. The manuscript focuses on not only the QC procedures but also how the two sets of data compare. The overall conclusions are that 1) levels of analytical precision are better (ie., lower CV) when samples are run ashore than at sea; 2) instrument responses are non-linear over the entire concentration range of the ocean from surface to depth and thus multiple CRMs are needed over this range; and 3) by better tracking of QC information in meta datafiles this will improve intercomparability of datasets generated by different people.

[Figure]

In summary this is an interesting paper, but really brings no new understanding to the body of knowledge.

Major concerns with the manuscript. 1. Only half the dataset is available. Unless I am missing something only the data from the MI analysis is present at the links given in the manuscript. The text suggests that, and looking closely at the XL file seems to confirm that. So actually this violates one of the primary review criteria of data accessibility. 2. How does this relatively unique comparison improve our ability to assess change in ocean nutrients over time? The authors conclude that having QC data available will improve intercomparability of at-sea generated datasets. It would have been nice to see an actual example of how they viewed this as working. As it is, I look at the control charts and see that there is IMMENSE day-to-day variability that is masked by the way they present the average CVs and RPDs in the paper. These control charts suggest that differences of 10% might need to happen between two datasets to say there is a change/difference between them. This seems rather large to me, especially for deep ocean nutrients. If one looks at ocean time-series, for example BATS, their deep nutrient data over 30 years hasn't changed anywhere near that much. 3. Focus of the recommendations for future work. As a nutrient chemist myself, the authors focus on recommendations that should be self-evident and thus seem to have limited value. Furthermore the recommendations focus on the idea that QC standards will 'fix' everything. Perhaps they will but the authors don't show that. Rather, I'm curious why the authors didn't make more recommendations about the importance of real procedural differences. Like splitting up your sample run into two (or more) components that are linear rather than trying to force a non-linear fitted function to the calibration data. I recognize that a single run is easier, but in the process are there compromises in data quality being made at both 'ends' (ie., shallow and deep). The authors try to demonstrate this in Table 5, so why not have a recommendation that in reprocessing data you only use the portion of the calibration curve around your data that is linear. I fear without focus on the actual methods, not just CRMS, that we will not get closer in our incomparability.

Minor concerns: 1. Given the other differences in Table 1, the authors should give development temperatures as well. Or if they are identical state as such. 2. Given that the authors label data as less than the LOQ or LOD in the datafile, perhaps including in the figures a dashed line representing the same values. 3. Perhaps I missed it, but the DAL method runs air through Cd column until reduction efficiency drops to 95%. How is the 5% loss of NO3 factored into the calculations? If it isn't now, would including it improve comparability given that 5% is one-half of the UAL/LAL range. 4. Line 454, I think they mean precision not accuracy. 5. Line 466-467. The authors comment on the difference between weighing standards and pipetting standards. Given they must know the error associated with both of those methods, could they not quantify that difference and thus provide more support for was is 'random' (and uncontrollable) error vs. error in controllable aspects of their methods. 6. Related to above. Line 469-470 – most of the random error (3% in the lab compared to 4-5% at sea) is not due to adverse conditions on the ship. Is this really random error? And if so what does that mean for detecting long term change in ocean nutrients. Why do some comparison exercises get 1%. The impacts of this on seeing change in the ocean need to be discussed. 7. Line 515: high quality shallow nutrient data is important. Geochemists use it to calculate parameters like N*. And thus our interpretation of ocean function is directly related to the quality of the measurements. This gets back to my earlier comment on recommendations that focus on methods as well as CRMs. 8. Table 1: concentrations of chemicals are needed not masses. 9. Table 3: a 1% measurement error is 2 decimal places. Can the values really be certified to 3 decimal places? 10. Figure 4: might be helpful to put a Zscore envelope around one of the analytical measurements to see where differences between datasets would be flagged 11. Figure 5: for the deeper data (>400m), in TOxN there is bias towards DAL measurements, but in PO4, it's a bias to MI. This isn't really discussed, is there a reason for this? 12. Some considerations for additional recommendations: a. Sample tube size, bigger tubes will minimize contamination effect during sampling. Is this also part of the difference between the two methods? b. Bottle type for freezing, see Dore et al. 1996 c. Is

Z-score =2 really OK or should we try to get it better?

---

## Referee Comment (RC3) · Anonymous Referee #3 · 9 Jun 2018

Review results of "A novel inter-comparison of nutrient analysis at sea: recommendations to enhance comparability of open ocean nutrient data" by McGrath et al.

In this paper, the authors reported the results, findings and lessons learned from a rare opportunity in which two independent nutrient analysis teams participated jointly in a deep ocean hydro graphic section A02. The reviewer completely agree with motivations of the authors to improve comparability of oceanic nutrients data and respect their effort as shown in this article. It is however that there are many issues which should be re-evaluated, re-calculated, re-check the fact about availability of CRMs, the treatment of certified values with uncertainty, comparison between measure value and certified

value and way of Z-scor calculation as stated below, the reviewer judges and request the authors to do major revision.

The major points:

Line 226-224 An Z score of 2 from QUASIMEME criteria is "acceptable" (line 231) in this article but the reviewer think that this criteria is not for deep ocean and Z-scor should be calculated using uncertainty of certification stated as K=2 for CRMs, not use Total error as the authors did in this article. In terms of SI tractability, we need to think about uncertainty of certified value. (Quasimeme criteria; the data is "satisfactory" if people do produce within a z of +/-2 at a taken proportional error of 6% and above that a constant error(offset) of e.g. 0.05umol/l for NO3) In coastal waters the variety can be large, like in the North Sea, it is however not at all comparable with open Ocean waters. If the reviewer calculates the max and minimum offset line within z=2 from Quasimeme for e.g. TOxN at the level of CD 5.5uM with the 6% prop error and the extra 0.05uM constant error the reviewer get a total error of (5.5*6%/100%)+(0.5*0.05)=0.355uM multiplied with 2 becomes +/-0.71um. So the limits are 5.5+/-0.71= 4.79 up to 6.21uM which give the level of "acceptance" if we use equation 2 for open ocean waters. The reviewer thinks that "4.79 up to 6.21uM" range is too big to accept for research at open oceans. If we use 2% as proportional error the limits will be: (5.5*2/100)+(0.5*0.05) *2=+/-0.27uM. Therefore we need to discuss our "satisfactory" level based on reality of variability of nutrients concentration in open water environment and capability of reproduciblility of nutrients measurements.. Therefore the reviewer asks the authors to revise this section to fit open water environment.

In addition, in line 231 z-score within 2 is mentioned as being "accepted" however Quasimeme use the term "satisfactory", see attachment page 9.(Round AQ1 2014-1) (and for the errors used page 13 Round 70 data).

Line311-313, Table 3 The reviewer found serious mistake about the certified value treatment. The certified concentration of KANSO CRMs are stated in micro mol kg-1,

not micro mol L-1, because in SI we need to use mol and kg. In Line 312 and table 3, Certified values are stated in micro mol L-1 but the concentrations in table 3 are same as certified values stated in micro mol kg-1. This mistake leads the authors about 2.5 % differences when the authors convert micro mol L-1 to micro mol kg-1 or micro mol kg-1 to micro mol L-1 using density of sample seawater or CRMs as appropriately. Please be careful that density of seawater depends sea water temperature and salinity as stated in TEOS10 (http://www.go-ship.org/Manual/TEOS-10_Manual_06Jul10.pdf).

Line330-366 The conclusion of the discussion in this article is good stated as "This firmly supports the recommendations of Hydes et al. (2010) concerning the importance of understanding and evaluating the best fit for an individual CFA system." It is however, the discussion here is not clear for the reviewer. Therefore the authors re-organize discussion here based on a view point of simple linearity/fitting method problem as discuses in current GO-SHIP nutrients manual.

Line 523-524 The authors stated that there is no low CRM available around the detection limit, but this is not true. Nutrients CRMs by KANSO, SCOR-JAMSTEC CRM and NMIJ CRMs all cover full range of nutrients concentration. There is batch BY(2015), SCOR-JAMSTC batch CE(2016, http://www.jamstec.go.jp/scor/available.html) and NMIJ CRM 7601a ( https://www.nmij.jp/english/service/C/crmlist_E_20180418.pdf) with all values close to detection.

Line 431-443 The authors stated that one of the key findings in this study is the need for using two (or more) reference materials for nutrient analysis that covers the range of the expected nutrients for the survey. Hydes et al. (2010) already recommends the use of CRMs to improve the comparability of the global ocean nutrient data set, and that a minimum of three reference material solutions (low, mid and top range) should be used at regular intervals during a cruise to detect non-linearity. It is obvious that we need to use at least three concentration levels CRMs from low, mid and high range. Theoretically two point CRM use is not enough as the authors faced in thier experiments. Therefore the authors reconsider this statement and should say more

appropriattly about use of CRM.

Line 486-496 The authors stated that we should mention a criteria for nutrients concentration in the GO-SHIP manual like other parameters. The reviewer agree with this, but before that all should state uncertainty with measured value using appropriate way and scientists need to study about magnitude of natural variability of nutrients concentration. After that our community can get good number of the criteria as appropriately.

Line 523-524 The authors stated that there is no low CRM available around the detection limit, but this is not true. Nutrients CRMs by KANSO, SCOR-JAMSTEC CRM and NMIJ CRMs all cover full range of nutrients concentration. There is batch BY(2015), SCOR-JAMSTC batch CE(2016, http://www.jamstec.go.jp/scor/available.html) and NMIJ CRM 7601a ( https://www.nmij.jp/english/service/C/crmlist_E_20180418.pdf) with all values close to detection.

Line 524-535 The reviewer observes that this part is a kind of conclusion or recommendation, therefore it might better to put conclusion/recommendation section

Minor points:

Page21 Table 5 In table 5, actually measured values for CRM CD and BW are in general high compared with certified values and they will be more close to the certified values if divided by approximately a density of 1.024 to convert from micro mol L-1 to micro mol kg-1. Also look at comment above.

Line 202-241 Procedure to prepare daily standards and treatment of blank value are unclear for the reviewer. Therefore it might better to add schematic diagram how the authors prepare daily working standards.

End of review.